# Non-Small-Cell Lung Cancer Immunotherapy and Sleep Characteristics: The Crossroad for Optimal Survival

**DOI:** 10.3390/diseases11010026

**Published:** 2023-02-01

**Authors:** Paul Zarogoulidis, Dimitrios Petridis, Christoforos Kosmidis, Konstantinos Sapalidis, Lila Nena, Dimitris Matthaios, Vasilis Papadopoulos, Eleni Isidora Perdikouri, Konstantinos Porpodis, Paschalis Kakavelas, Paschalis Steiropoulos

**Affiliations:** 1Pulmonary Department, General Clinic Euromedica Private Hospital, 68100 Thessaloniki, Greece; 23rd Surgery Department, AHEPA University Hospital, Aristotle University of Thessaloniki, 54646 Thessaloniki, Greece; 3Department of Food Technology, School of Food Technology and Nutrition, Alexander Technological Educational Institute, 64556 Thessaloniki, Greece; 4Laboratory of Social Medicine, Medical School, Democritus University of Thrace, 68100 Alexandroupolis, Greece; 5Oncology Department, General Hospital of Rhodes, 85100 Rhodes, Greece; 6Oncology Department, University Hospital of Larissa, 77543 Thessali, Greece; 7Oncology Department, General Hospital of Volos, 34456 Volos, Greece; 8Pulmonary Department, “G. Papanikolaou” General Hospital, Aristotle University of Thessaloniki, 54768 Thessaloniki, Greece; 9Intensive Care Unit, General Clinic Euromedica, Private Hospital, 54667 Thessaloniki, Greece; 10Department of Respiratory Medicine, Medical School, Democritus University of Thrace, 68100 Alexandroupolis, Greece

**Keywords:** non-small-cell lung cancer, immunotherapy, sleep medicine, polysomnography, cancer, insomnia

## Abstract

Introduction: Non-small-cell lung cancer is still diagnosed at an inoperable stage and systematic treatment is the only option. Immunotherapy is currently considered to be the tip of the arrow as the first-line treatment for patients with a programmed death-ligand 1 ≥ 50. Sleep is known to be an essential part of our everyday life. Patients and Methods: We investigated, upon diagnosis and after nine months, 49 non-small-cell lung cancer patients undergoing immunotherapy treatment with nivolumab and pemprolisumab. A polysomnographic examination was conducted. Moreover, the patients completed the Epworth Sleepiness Scale (ESS), the Pittsburgh Sleep Quality Index (PSQI), the Fatigue Severity Scale (FSS) and the Medical Research Council (MRC) dyspnea scale. Results: Tukey mean-difference plots, summary statistics, and the results of paired *t*-test of five questionnaire responses in accordance with the PD-L1 test across groups were examined. The results indicated that, upon diagnosis, patients had sleep disturbances which were not associated with brain metastases or their PD-L1 expression status. However, the PD-L1 status and disease control were strongly associated, since a PD-L1 ≥80 improved the disease status within the first 4 months. All data from the sleep questionnaires and polysomnography reports indicated that the majority of patients with a partial response and complete response had their initial sleep disturbances improved. There was no connection between nivolumab or pembrolisumab and sleep disturbances. Conclusion: Upon diagnosis, lung cancer patients have sleep disorders such as anxiety, early morning wakening, late sleep onset, prolonged nocturnal waking periods, daytime sleepiness, and unrefreshing sleep. However, these symptoms tend to improve very quickly for patients with a PD-L1 expression ≥80, because disease status improves also very quickly within the first 4 months of treatment.

## 1. Introduction

Lung cancer patients are still diagnosed at an advanced, inoperable stage. An effort is currently being made by the pneumonology community to inform, educate, and urge people at a high risk of lung cancer to perform computed tomography scans as a method of early lung cancer detection [1]. We have novel techniques for a safe biopsy in small pulmonary nodules with radial-ebus and electromagnetic navigation systems [2,3,4,5,6]. Unfortunately, the diagnosis of lung cancer is made at an advanced, inoperable stage. It has been previously observed that these patients have severe sleep disturbances, which are associated with a poor quality of life [7]. The usual sleep disorders are anxiety, early morning wakening, late sleep onset, prolonged nocturnal waking periods, daytime sleepiness, and unrefreshing sleep [8]. It has been observed that insomnia symptoms have a variation between hospitalizations and between cancer treatments and side effects [9,10,11]. Unfortunately, sleep disturbances increase symptoms such as pain, fatigue, and depression [12,13]. In newly diagnosed cancer patients, sleep disturbances have been reported in up to 50% of patients for all cancer types. The importance of sleep disturbances has been underestimated [14]. In most studies with sleep disturbances and cancer patients, different types of cancer were included, usually breast cancer and prostate cancer [9] There are few studies published with lung cancer patients receiving treatment with chemotherapy regimens [15,16]. In our present pilot study, we evaluated a subgroup of lung cancer, non-small-cell lung cancer (NSCLC) patients, who exhibit a programmed death-ligand 1 ≥ 50. This is a special group in which we administer immunotherapy alone as a first-line treatment [17]. These patients have a different survival profile and restaging methodology [18]. Until ten years ago, we only had chemotherapy regimens and tyrosine kinase inhibitors (TKIs). In the past ten years, the novel treatment of immunotherapy was introduced with different medications. Moreover, combinations of chemotherapy and immunotherapy are administered [19]. In our pilot study, we evaluated this specific subgroup of NSCLC patients using polysomnography and sleep questionnaires. We report the characteristics of these patients upon diagnosis without treatment and the impact of treatment after the second re-staging.

## 2. Patients and Methods

Forty-nine patients (forty-four males and five females), with primary lung cancer diagnoses were included in the pilot study. All patients had non-small-cell lung cancer (NSCLC), either adenocarcinoma or squamous cell carcinoma; we did not include non-other specific (NOS) types in order to maintain a homogenous genetic sample. In total, 14 were Stage IIIb and 35 were Stage IV. We obtained demographic and clinical data regarding age, sex, painkillers, brain metastasis, and we finally recorded the Eastern Cooperative Oncology Group (ECOG) performance status. All patients included had programmed death-ligand 1 (PD-L1) status ≥ 50% and were stratified into three categories: 4 = 50–70, 5 = 71–90, and 6 = 91–100 in our statistics. Based on previous studies, a higher expression implies that survival is prolonged. Since they had a PD-L1 ≥ 50%, all patients received only immunotherapy, either with pembrolisumab or nivoslumab as the first-line treatment. We did not include patients receiving corticosteroids orally or intravenously, however, we did include patients with chronic obstructive pulmonary disease (COPD) under treatment with inhaled corticosteroids. All patients included did have chronic heart failure, NIHA ≤ III, or they had their heart disease under control. The main exclusion criteria included the inability to understand and answer the questionnaires that were distributed for their sleep evaluation (see next section). The study was approved by our IRB 29/2022 the “AHEPA” hospital. Written informed consent was obtained from each patient before study enrollment.

## 3. Sleep Evaluation Methodology

Patients completed self-report questionnaires one day before their first treatment and again at nine months after. In order to evaluate daytime sleepiness, we used the Greek version of the Epworth Sleepiness Scale (ESS) [20]. We used the ESS to assess daytime sleepiness over the last three months under eight usual circumstances. The ESS was evaluated for the Greek population and the cut-off point, indicating excessive daytime sleepiness, was set at 10. We used the Greek version of the Pittsburgh Sleep Quality Index (PSQI) in order to assess the sleep quality [21]. The PSQI is composed of 19 self-rated questions that are grouped into seven domains (subjective sleep quality [SSQ], sleep latency [SL], sleep duration [SDU], habitual sleep efficiency [HSE], sleep disturbances [SDI], use of sleeping medication [SM], and daytime dysfunction [DD]). In order to evaluate fatigue, we used the Greek version of the Fatigue Severity Scale (FSS) [22]. In order to objectively evaluate sleep quality, we used overnight polysomnography (PSG) (Alice 3, Respironics) with a standard montage of electroencephalogram (EEG), electrooculogram, electromyogram (EMG), and electrocardiogram (ECG) signals in combination with pulse oximetry and airflow, which were detected using combined oronasal thermistors. The thoracic cage and abdominal motion were recorded by inductive plethysmography. EEG recordings were manually scored according to standard criteria. We used the Medical Research Council (MRC) scale in order to evaluate the presence and grade of dyspnea [23].

## 4. Statistics and Results

We employed Tukey mean-difference plots, summary statistics, and the results of the paired *t*-test of five questionnaire responses in accordance with the PD-L1 test across groups. Horizontal and vertical lines determined the mean difference, plus the 95% confidence intervals and the mean of paired sums, accordingly (Appendix A).

The obstructive sleep apnea syndrome was investigated via the responses derived the disease questionnaires and recorded at two consecutive time intervals, before and after immunotherapy treatment, in 49 patients. The matched-pairs statistical criterion was adopted because it compares individual differences and the means of sums using the paired *t*-test. The Tukey mean-difference plot was also employed graphically. Additionally, the effect of PD-L1 expression in the repeated two periods was tested across the groups, coded as 4 (50–70%), 5 (71–90%), and 6 (91–100%). Pain was coded as 1 (present) and 2 (absent), thus producing four combined categories for the two consecutive periods: 11, 12, 21, and 22.

Forty-nine patients, averaging 61 years old, mostly males (89.8%, Table 1 and Table 2), and ranging for the 50% between 53.5 and 69 y.o., comprised the whole population (Figure 1).

According to Table 1, cancer at Stages IIIb and IV was encountered nearly equally (47% and 43%). Adenocarcinoma was observed in a 2:1 proportion to squamous, while patients with no NSCLC BRAIN METS and no radiation exposure were recorded in the proportion of 3:1. Eleven patients suffered continuous pain in both time periods, twelve recuperated, and thirty six never suffered. Regarding the BMI condition, 22 individuals were overweight, 16 were normal, and 11 were obese. The PD-L1 expression was expectedly found in very good agreement with the disease progress (96.3%), as Figure 2 vividly demonstrates, showing only three mismatches and one outlier (PD).

No significant effect of any questionnaire response was found between the time intervals (before and after treatment with immunotherapy), as Appendix A shows in detail for the paired *t*-test results. A significant effect of PD-L1 was not statistically documented (*p*-values greater than 0.05) across the expression groups, concerning either mean differences or the means of mean sums between pairs’ responses.

## 5. Discussion

All newly diagnosed patients had poor sleep quality, which was associated with daily fatigue. As previously observed in other immunotherapy studies, patients with high a PD-L1 status (≥80%) have a faster response and a prolonged response [24]. We stratified our patients, as in previous studies, into three categories in order to perform our analysis [25,26]. It was observed that patients in Groups 5 and 6, meaning ≥71% PD-L1 expression, indeed had faster responses and complete responses; hence, their sleep disorder symptoms were alleviated. Upon diagnoses, the most affected component was sleep latency, followed by sleep duration. During therapy, all patients who were using painkillers discontinued their use due to the efficiency of the therapy. Dyspnea is a major symptom for patients with chronic obstruction pulmonary disease (COPD); it affects both daytime function and sleep quality. However, in our study, all not all patients were smokers. Additionally, the mean MRC score upon diagnosis was low. Moreover, no relationship was observed between the MRC score and sleep quality.

Upon diagnosis, the reduced sleep duration was connected to poor sleep quality, and insomnia was the most common sleep disturbance observed in these lung cancer patients, with a major impact on their quality of life. In contrast to our study, in previous studies that investigated immunotherapy, insomnia remained even in patients who exhibited a partial response or a complete response [27]. Moreover, the sleep disturbance symptoms in these studies were indifferent to the PD-L1 expression status. Mild daytime sleepiness was reported, along with poor sleep duration and efficiency. In our study, no correlation was observed between daytime sleepiness and sleep quality, as assessed by the global PSQI score or any of its components. Fatigue was observed in our patients, which was associated with sleep quality, subjective sleep quality, sleep duration, and daily dysfunction. These findings are in accordance with previous studies [21,28,29,30]. After nine months of immunotherapy, these correlations changed dramatically, and all symptoms for those patients with a complete response were alleviated almost completely. It is known that cancer-related fatigue is a common symptom, and its prevalence reaches 80 to 90% of cancer patients under treatment [31,32]. Fatigue symptoms are often under-diagnosed and under-treated by health care practitioners. However, fatigue has a huge impact on the patients’ quality of life and daytime activities. Fatigue has an effect on the central nervous system, muscle metabolism, circadian rhythm, inflammatory and stress mediators, immune system activation, and hormonal alteration [29]. Therefore, the impact of treatment in these patients has a huge effect, as their effective therapy (treatment response) reduced the stress hormones (cortisol) and activated the immune system. We did not observe any differences between the two drugs, nivolumab or pembrolisumab. Patients with a partial response and mostly those with a complete response experienced a positive effect on ECOG status, pain status, and psychology status, with an improved sleep quality and reduced fatigue and insomnia, which again enhanced the treatment and response. The limitations of the present study are its small sample size and the small number of women in the sample. Furthermore, we did not keep record of the number of women in peri-menopausal or menopausal age, or the drugs that they were taking for these purposes. Women receiving drug medication for these conditions tend to have their symptoms ameliorated. The mean age of the five women in the study was 62 years of age. Additionally, different hormone levels were not measured to observe the impact and interaction of immunotherapy and body environment. Moreover, future studies should compare more patients with a PD-L1 status ≤50%; this category of patients receive chemotherapy along with immunotherapy.

## 6. Conclusions

The sleep quality of lung cancer patients, subjectively evaluated, was poor upon diagnosis. Sleep efficiency, objectively measured, increased during the course of immunotherapy for patients in all groups, but mostly for patients with a PD-L1 expression ≥91–100%. Fatigue was the major symptom associated with sleep quality in these patients upon diagnosis, and the first to be alleviated during treatment. Sleep quality in patients with lung cancer should be evaluated in large-scale, prospective studies, including PSG, in order to obtain a better understanding of the mechanisms of poor sleep and its contribution to daily fatigue. We should bear in mind that sleep impairment should be also one of our treatment targets, since immunotherapy is associated with hormones and PD-L1 status.

## Figures and Tables

**Figure 1 diseases-11-00026-f001:**
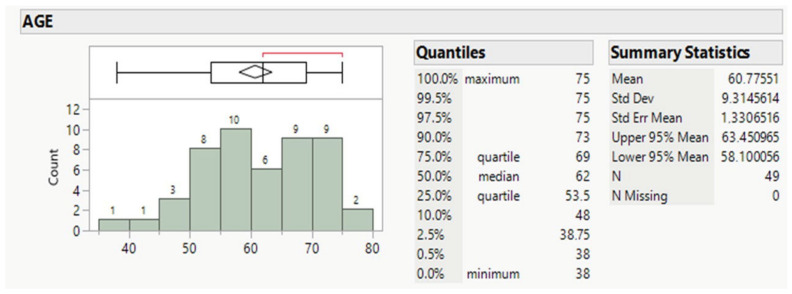
Age distribution of patients.

**Figure 2 diseases-11-00026-f002:**
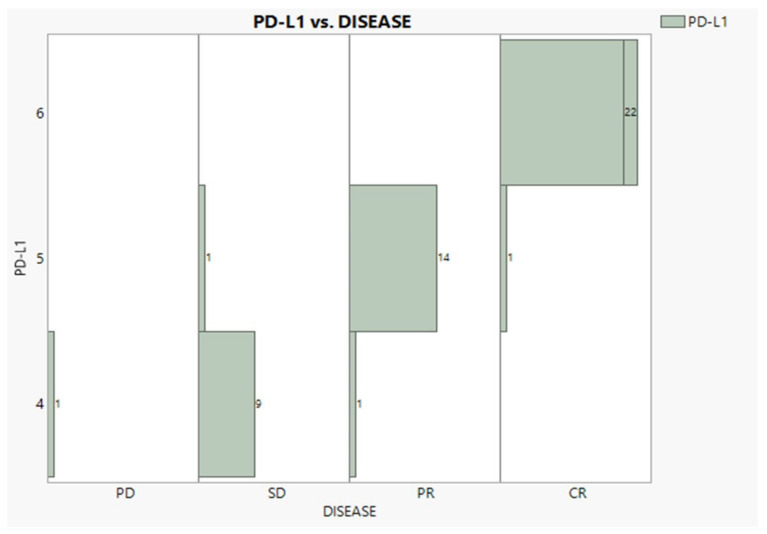
Relationships between PD-L1 expression and disease progress.

**Table 1 diseases-11-00026-t001:** Basic features of the study.

Sex	N	% of Total
Female	5	10.20%
Male	44	89.80%
**Stage**		
IIIb	26	53.06%
IV	23	46.94%
**Histology**		
Adeno	32	65.31%
Squamous	17	34.69%
**Brain**		
Brain METS	12	24.49%
No Brain METS	37	75.51%
**Radiation**		
NO	37	75.51%
YES	12	24.49%
**PD-L1**		
4	11	22.45%
5	16	32.65%
6	22	44.90%
**Pain Combined**		
11	8	16.33%
12	5	10.20%
22	36	73.47%
**Disease**		
PD	1	2.04%
SD	10	20.41%
PR	15	30.61%
CR	23	46.94%
**BMIi**		
Normal	16	32.65%
Overwight	22	44.90%
Obese	11	22.45%

**Table 2 diseases-11-00026-t002:** Descriptive statistics of the questionnaire responses in the two consecutive time intervals.

Questionaire	Min	Max	Mean	Std Dev	Median	25% Quartile	75% Quartile	Interquartile
AHΙ	0	114.2	22.381633	29.314769	9.8	1.75	33.3	31.55
FSS	7	63	33.897959	17.986389	31	21.5	43.5	22
ATHENS	0	18	5.4489796	4.8822008	4	1.5	8	6.5
PITBURGH	1	17	7.1632653	4.012205	6	4.5	9	4.5
EPWORTH	0	15	6.8163265	4.3954212	7	3	10	7
AHΙ 2	0	471	27.657143	70.82073	3.9	1.55	26.55	25
FSS 2	9	64	37.755102	19.763741	39	16	56	40
ATHENS 2	0	24	3.7959184	6.3310728	1	0	4	4
PITBURGH 2	0	17	6.2244898	4.8445214	4	2	9	7
EPWORTH 2	0	15	5.6938776	4.302151	5	2	9	7

## Data Availability

Any data can be provided by the corresponding author if requested.

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
