# Peer review of "Non-Small-Cell Lung Cancer Immunotherapy and Sleep Characteristics: The Crossroad for Optimal Survival"

_diseases, 2023, doi:10.3390/diseases11010026_

Round 1

Reviewer 1 Report

The study is of interest because of its attempt to examine the relationship between sleep characteristics and NSCLC immunotherapy.

there are minor English grammar errors that should be picked up by the editing process

I believe that there are several issues that need to be addressed prior to publication.

1. given the age distribution, normal sleep patterns would be expected to very thus presenting a challenge to the measurement,  analysis and interpretation that should be addressed

2. although specific ages for women included in the study were not given, it is possible that some  may have been peri-menopausal or menopausal in which case both sleep patterns may be perturbed and other medications may be used to ameliorate symptoms associated with the transition.  These might impact the sleep patterns as well independent of the immunotherpy

3. in the general population, issues of co-morbid conditions and poly-pharmacy should be considered, evaluated discussed in terms of the potential impact

Author Response

Reviewer 1

The study is of interest because of its attempt to examine the relationship between sleep characteristics and NSCLC immunotherapy.

there are minor English grammar errors that should be picked up by the editing process

Answer

Thank you for your comments, yes we have made the necessary corrections which have been highlighted in yellow

I believe that there are several issues that need to be addressed prior to publication.

  1. given the age distribution, normal sleep patterns would be expected to very thus presenting a challenge to the measurement,  analysis and interpretation that should be addressed.

Answer

Thank you for your comment

Indeed there were several issues, first of all in Greece patients underestimate the importance of sleep disturbances and its impact in their treatment schedule and treatment effectiveness. Hopefully we were able to convince several of these patients to evaluate and re-evaluate their sleep. Moreover; the sleep patterns upon diagnosis did not differ among patients since they were patients with the same disease stage and all had PD-L1 ≥50%, this is one explanation. Minor differences in sleep patterns were observed after nine months of treatment among those with partial response and complete response and no –responders. Which is expected since in non-responders the symptoms along with the rest of the physical status worsens. The small sample of our study did not allow use to have more data to compare among responders and non-responders. However; we believe that even in a larger sample we would observe again that sleep disturbances would be alleviated in the responders, since the physical status would be improved. Mayde a study with a longer time frame like 2 years would give us more data.

  1. although specific ages for women included in the study were not given, it is possible that some  may have been peri-menopausal or menopausal in which case both sleep patterns may be perturbed and other medications may be used to ameliorate symptoms associated with the transition.  These might impact the sleep patterns as well independent of the immunotherapy

Answer

Thank you for your comment

This is a limitation of our study. We have mentioned the mean age of the women population in our text (discussion section), however; we did not kept record of such medication, this is now mentioned in our text.

  1. in the general population, issues of co-morbid conditions and poly-pharmacy should be considered, evaluated discussed in terms of the potential impact

Answer

Thank you for your comment

You are right, and therefore we did not include patients taking cortisone and we did not have patients with pneumonitis or thyroiditis. This is now mentioned in the inclusion criteria and also all patients had their copd and heart condition under control when they initiated their treatment. Moreover; this specific group of patients with PD-L1≥50% usually tend to respond to treatment and therefore their situation only gets better and not worse at least for the first year.

Reviewer 2 Report

1. In the study, sleep characteristics, its relationship with PD-L1 level and the change in sleep quality according to PD-L1 level after immunotherapy were evaluated in patients with stage 3 and 4 NSCLC. It has been emphasized that there is a rapid improvement in sleep quality in patients with high PD-L1 levels, and this can already be predicted by the treatment response. Therefore, studies have shown that patients with already high PD-L1 levels have a better response to immunotherapy and respond quickly to treatment. In this context, it has not been evaluated as a study that can have clinical reflection.

2. The literature has been evaluated, but it is seen that no meaningful study design can be made.

3. The study was not well designed and its integration into the clinic does not seem possible.

4.  It is seen that the results are presented clearly.

5. The effects and stages of the framework carried out are not well designed. It is not a meaningful study that can be reflected in daily practice.

6. The communication quality of the paper is clear. The authors state what they do well, but overall, the proposed hypothesis is not promising and clinically meaningless. The language of the article is not good and contains typos as far as I can detect.

Author Response

Reviewer 2

The communication quality of the paper is clear. Authors state what they do well, however, overall, the proposed hypothesis is not promising and clinically meaningless.

Answer                                           

Thank you for your comment

We respect your opinion, but we disagree, below are the answers to your comments.

  1. In the study, sleep characteristics, its relationship with PD-L1 level and the change in sleep quality according to PD-L1 level after immunotherapy were evaluated in patients with stage 3 and 4 NSCLC. It has been emphasized that there is a rapid improvement in sleep quality in patients with high PD-L1 levels, and this can already be predicted by the treatment response. Therefore, studies have shown that patients with already high PD-L1 levels have a better response to immunotherapy and respond quickly to treatment. In this context, it has not been evaluated as a study that can have clinical reflection.

Answer

Thank you for your comment

We disagree

Given the age distribution, normal sleep patterns would be expected to vary thus presenting a challenge to the measurement, analysis and interpretation that should be addressed.

Indeed there were several issues, first of all in Greece patients underestimate the importance of sleep disturbances and its impact in their treatment schedule and treatment effectiveness. Hopefully we were able to convince several of these patients to evaluate and re-evaluate their sleep. Moreover; the sleep patterns upon diagnosis did not differ among patients since they were patients with the same disease stage and all had PD-L1 ≥50%, this is one explanation. Minor differences in sleep patterns were observed after nine months of treatment among those with partial response and complete response and no –responders. Which is expected since in non-responders the symptoms along with the rest of the physical status worsens. The small sample of our study did not allow use to have more data to compare among responders and non-responders. However; we believe that even in a larger sample we would observe again that sleep disturbances would be alleviated in the responders, since the physical status would be improved. Mayde a study with a longer time frame like 2 years would give us more data.

  1. The literature has been evaluated, but it is seen that no meaningful study design can be made.
    Answer

Thank you for your comment, however; we do not agree. Our study design is according to previously published studies and it is a mix of evaluating lung cancer patients and sleep disturbances

  1. The study was not well designed and its integration into the clinic does not seem possible.

Answer

Thank you for your comment, however; we do not agree. Our study design is according to previously published studies and it is a mix of evaluating lung cancer patients and sleep disturbances

  1. It is seen that the results are presented clearly.

Answer

Thank you

5. The effects and stages of the framework carried out are not well designed. It is not a meaningful study that can be reflected in daily practice.
Answer

Thank you for your comment, however; we do not agree. Our study design is according to previously published studies and it is a mix of evaluating lung cancer patients and sleep disturbances

Also, sleep disturbances in lung cancer patients is a complex issue and these patients should be observed only in centers with both oncology and pulmonary departments.

  1. The communication quality of the paper is clear. The authors state what they do well, but overall, the proposed hypothesis is not promising and clinically meaningless. The language of the article is not good and contains typos as far as I can detect.

Answer

Overall sample size for the current study is low, thus the results might not be reliable given inadequate power of the study. This issue has been mentioned in the discussion section as a major limitation of the study. Regarding the priori power calculation our statistician disagrees as it is his belief that valid conclusions can be concluded with the statistics methodology that is currently presented.

Thank you for your comment, we have now corrected the manuscript both grammatically and linguistically

Reviewer 3 Report

The study by Zarogoulidis et al. evaluated the NSCLC patients with polysomnography and sleep questionnaires. The results indicated that patients upon diagnosis had sleep disturbances which were not associated with brain metastases or PD-L1 expression status. However; the PD-L1 status and disease control were strongly associated, since PD-L1 ≥80 improved the disease status within the first 4 months. Following are some concerns which need to be addressed by the authors.

1.     Overall sample size for the current study is low, thus the results might not be reliable given inadequate power of the study. An a priori power calculation would have been useful.

2.     Check grammatically and spelling throughout the manuscript. There are some mistakes.

3.     The References should be checked carefully, for example, References 2, 5, 6, 8, 17 and 19 should been revised.

Author Response

Reviewer 3

The study by Zarogoulidis et al. evaluated the NSCLC patients with polysomnography and sleep questionnaires. The results indicated that patients upon diagnosis had sleep disturbances which were not associated with brain metastases or PD-L1 expression status. However; the PD-L1 status and disease control were strongly associated, since PD-L1 ≥80 improved the disease status within the first 4 months. Following are some concerns which need to be addressed by the authors.

  1. Overall sample size for the current study is low, thus the results might not be reliable given inadequate power of the study. An a priori power calculation would have been useful.

Answer

Thank you for your comment

You are right and this issue has been mentioned in the discussion section as a major limitation of the study. Regarding the priori power calculation our statistician disagrees as it is his belief that valid conclusions can be concluded with the statistics methodology that is currently presented.

  1. Check grammatically and spelling throughout the manuscript. There are some mistakes.

Answer

Thank you for your comment, we have now corrected the manuscript both grammatically and linguistically

  1. The References should be checked carefully, for example, References 2, 5, 6, 8, 17 and 19 should been revised.

Answer

Thank you for your comment, however; we do not understand exactly what you mean since we used the reference program endnote with the reference style ACS as indicated in the author guidelines.

Reviewer 4 Report

Paul Zarogoulidis and colleagues present a quality and well-written communication manuscript focused on non-small cell lung cancer immunotherapy and sleep characteristics with respect to the crossroad for optimal survival.

Authors investigated upon diagnosis and after nine months, 49 non-small cell lung cancer patients undergoing immunotherapy treatment with nivolumab and pemprolisumab a polysomnographic examination was conducted and the patients completed the Epworth Sleepiness Scale, the Pittsburgh Sleep Quality Index, the Fatigue Severity Scale and the Medical Research Council dyspnea scale. 

Authors performed Tukey mean-difference plots, summary statistics and results of paired t-test of five questionnaire responses in accordance with PD-L1 test across groups. The results indicated that patients upon diagnosis had sleep disturbances which were not associated with brain metastases or PD-L1 expression status. However; the PD-L1 status and disease control were strongly associated, since PD-L1 ≥80 improved the disease status within the first 4 months. All data from sleep questionnaires and polysomnography reports indicated that the majority of patients with partial response and complete response had their initial sleep disturbances improved. There was no connection between nivolumab or pembrolisumab and sleep disturbances. 

Finally, authors conclude that lung cancer patients upon diagnosis have sleep disorders such as anxiety, early morning wakening, late sleep onset, prolonged nocturnal waking periods, daytime sleepiness and unrefreshing sleep. However; these symptoms tend to improve very quickly for patients with PD-L1 expression ≥80, because disease status improves also very quickly within the first 4 months of treatment. They also add that sleep quality in patients with lung cancer should be evaluated in large-scale, prospective studies, including PSG, in order to obtain a better understanding of the mechanisms of poor sleep and its contribution to daily fatigue. Also it should be beared in mind that since immunotherapy is associated with hormones and PD-L1 status, sleep impairment should be also one of our treatment targets.

Overall, the manuscript is valuable for the scientific community and should be accepted for publication after edits are made.

===========================

Other comments:

1) Please check for typos throughout the manuscript.

2) With regards to PD-L1 expression and targeting – authors are kindly encouraged to cite the following article that describes novel immunotherapeutics targeting PD-L1 and its signaling pathway.
DOI: 10.3390/cancers14225539

Author Response

Reviewer 4

Paul Zarogoulidis and colleagues present a quality and well-written communication manuscript focused on non-small cell lung cancer immunotherapy and sleep characteristics with respect to the crossroad for optimal survival.

Authors investigated upon diagnosis and after nine months, 49 non-small cell lung cancer patients undergoing immunotherapy treatment with nivolumab and pemprolisumab a polysomnographic examination was conducted and the patients completed the Epworth Sleepiness Scale, the Pittsburgh Sleep Quality Index, the Fatigue Severity Scale and the Medical Research Council dyspnea scale. 

Authors performed Tukey mean-difference plots, summary statistics and results of paired t-test of five questionnaire responses in accordance with PD-L1 test across groups. The results indicated that patients upon diagnosis had sleep disturbances which were not associated with brain metastases or PD-L1 expression status. However; the PD-L1 status and disease control were strongly associated, since PD-L1 ≥80 improved the disease status within the first 4 months. All data from sleep questionnaires and polysomnography reports indicated that the majority of patients with partial response and complete response had their initial sleep disturbances improved. There was no connection between nivolumab or pembrolisumab and sleep disturbances. 

Finally, authors conclude that lung cancer patients upon diagnosis have sleep disorders such as anxiety, early morning wakening, late sleep onset, prolonged nocturnal waking periods, daytime sleepiness and unrefreshing sleep. However; these symptoms tend to improve very quickly for patients with PD-L1 expression ≥80, because disease status improves also very quickly within the first 4 months of treatment. They also add that sleep quality in patients with lung cancer should be evaluated in large-scale, prospective studies, including PSG, in order to obtain a better understanding of the mechanisms of poor sleep and its contribution to daily fatigue. Also it should be beared in mind that since immunotherapy is associated with hormones and PD-L1 status, sleep impairment should be also one of our treatment targets.

Overall, the manuscript is valuable for the scientific community and should be accepted for publication after edits are made.

===========================

Other comments:

  • Please check for typos throughout the manuscript.

Answer

Thank you for your comments, we made the necessary corrections throughout the text which have been highlighted in yellow

  • With regards to PD-L1 expression and targeting – authors are kindly encouraged to cite the following article that describes novel immunotherapeutics targeting PD-L1 and its signaling pathway.
    DOI: 10.3390/cancers14225539

Answer

Thank you we have now added the referenced

Round 2

Reviewer 1 Report

authors addressed my concerns

Author Response

Reviewer 1

Thank you

Reviewer 3 Report

Accept in present form

Author Response

Reviewer 3

Thank you
